# Development syndromes in New World temperate and tropical songbirds

**Suzanne H. Austin**[1,2,3]*, **W. Douglas Robinson**[1], **Tara Rodden Robinson**[1], **Vincenzo A. Ellis**[4,5], **Robert E. Ricklefs**[2]

1 Department of Fisheries and Wildlife, Oregon State University, Corvallis, Oregon, United States of America,
2 Department of Biology, University of Missouri-St. Louis, St. Louis, Missouri, United States of America,
3 Department of Integrative Biology, Oregon State University, Corvallis, Oregon, United States of America,
4 Department of Biology, Lund University, Lund, Sweden, 5 Department of Entomology and Wildlife Ecology, University of Delaware, Newark, Delaware, United States of America

* suzannehaustin@gmail.com

## Abstract

We studied avian development in 49 to 153 species of temperate and tropical New World passerine birds to determine how growth rates, and incubation and nestling periods, varied in relation to other life-history traits. We collected growth data and generated unbiased mass and tarsus growth rate estimates (mass $n$ = 92 species, tarsus $n$ = 49 species), and measured incubation period ($n$ = 151) and nestling period ($n$ = 153), which we analyzed with respect to region, egg mass, adult mass, clutch size, parental care type, nest type, daily nest predation rate (DMR), and nest height. We investigated covariation of life-history and natural-history attributes with the four development traits after controlling for phylogeny. Species in our lowland tropical sample grew 20% (incubation period), 25% (mass growth rate), and 26% (tarsus growth rate) more slowly than in our temperate sample. Nestling period did not vary with respect to latitude, which suggests that tropical songbirds fledge in a less well-developed state than temperate species. Suboscine species typically exhibited slower embryonic and post-embryonic growth than oscine passerines regardless of their breeding region. This pattern of slow development in tropical species could reflect phylogenetic effects based on unknown physiological attributes. Time-dependent nest mortality was unrelated to nestling mass growth rate, tarsus growth rate, and incubation period, but was significantly associated with nestling period. This suggests that nest predation, the predominant cause of nest loss in songbirds, does not exert strong selection on physiologically constrained traits, such as embryonic and post-embryonic growth, among our samples of temperate and lowland tropical songbird species. Nestling period, which is evolutionarily more labile than growth rate, was significantly shorter in birds exposed to higher rates of nest loss and nesting at lower heights, among other traits. Differences in life-history variation across latitudes provide insight into how unique ecological characteristics of each region influence physiological processes of passerines, and thus, how they can shape the evolution of life histories. While development traits clearly vary with respect to latitude, trait distributions overlap broadly. Life-history and natural history associations differ for each development trait, which suggests that unique selective pressures or constraints influence the evolution of each trait.

**Data Availability Statement:** Austin, S.H., Robinson, W.D., Robinson, T.R., Ricklefs, R.E. (2020). Development syndromes in New World temperate and tropical songbirds (Version 1) [Data

set]. Oregon State University. https://doi.org/10.7267/m613n4425.

**Funding:** Project funding was provided by National Science Foundation IRCEB grant #0212587 to RER and WDR. VAE was supported by a postdoctoral fellowship from the Carl Tryggers Foundation. The funders had no role in study design, data collection and analysis, decision to publish, or preparation of the manuscript.

**Competing interests:** The authors have declared that no competing interests exist.

## Introduction

Life-history theory presupposes that organisms optimize fitness trade-offs to optimally balance reproduction and survival [1]. These trade-offs may create axes of variation among life-history traits that may vary across latitude [2,3], reflecting environments that favor different pace-of-life syndromes. The slower pace-of-life of many lowland tropical birds compared to temperate species is associated with their higher rate of adult survival and lower annual reproductive rate [4–8]. This apparent fitness trade-off suggests that tropical birds engage in a strategy of increased investment in fewer individual offspring, potentially leading to higher quality and competitively advantaged fledglings [7,9–11]. Meanwhile, temperate songbirds typically invest in a larger number of offspring, which suffer higher rates of mortality [12] and lower recruitment [13]. Traits associated with higher investment in individual offspring of tropical birds include larger eggs relative to body size [14], smaller clutches [15], longer incubation periods [7], slower growth rates [7,9], and longer periods of post-fledging care [11]. A general trend toward lower annual reproductive success is also associated with tropical species [14].

Development traits vary conspicuously with latitude, reflecting the slower pace-of-life phenotype apparently favored in the tropics. Small (< 100 g), lowland tropical songbirds have, on average, 10% longer incubation periods and 23% slower growth rates than temperate passerines [7]. Several hypotheses address this difference in development times. One suggests that slower development allows increased investment in immune function [16,17]. The positive correlation between nestling growth rate and the period required to produce a successful brood may limit fecundity by limiting the time available to produce more broods in a season [4,18,19]. If season length exerts a primary environmental constraint on fecundity, then selection should favor faster development in north temperate regions where the short breeding season, and the time parents require to rear a successful brood, limits reproductive rate. There, selection should favor shorter development periods (and faster growth) to optimize offspring fitness and time to independence with respect to parental fecundity and opportunities to re-nest.

Nest predation has also been postulated as a primary driver of nestling growth rate and duration of development. All else being equal, selection should favor shorter incubation or nestling periods to reduce mortality risk of embryos and nestlings [20–22]. However, support for the role of nest predation in driving variation in embryonic and post-embryonic development periods is inconsistent across studies. Nest predation rate varies spatially and temporally, whereas growth rate is physiologically constrained and evidently less evolutionarily labile [10,19]. The level of support for different hypotheses is also influenced by the method used to estimate growth rates [23].

Nest predation rate is, on average, higher in the tropics, but its distribution shows extensive overlap comparing tropical and temperate regions [10,24–26]. Predation might influence parental nest attendance and feeding behavior, as parents seek to reduce perceived predation risk to themselves and to their offspring by limiting their activity around the nest site [20,27–30]. Lower parental attendance could influence development rates by reducing incubation temperatures and, later, the rate of food provisioning to the nestlings, thereby prolonging development time. Thus, predation risk could indirectly affect development via its influence on parental care behavior. Parental attendance (uni- vs. bi-parental care) also could influence the amount of care provided to eggs and nestlings, which, in turn, could influence embryonic and post-embryonic development [30]. Thus, lower attendance, often associated with female-only incubation and parental care, might limit growth by limiting temperature, or food resources, for optimal development [20,30].

Other natural-history traits, including nest type and nest height, might influence development indirectly by their impact on nest predation. Different nest types are associated with different levels of nest predation [10,18,28,31,32]. Effects of egg and nestling mortality might be mitigated by nest height, as nests placed closer to the ground tend to have higher daily nest mortality rate (DMR), and by nest type, as cavity and enclosed-cup nests tend to have lower DMR [10,33–35]. However, nestling period, not nestling growth rate, is thought to be more responsive to nest predation rate [18,19,but see 21], because growth rate is additionally influenced by functional and structural maturity of the developing chick [19], the quantity and quality of resources that parents provide, and the rates at which nutrients are assimilated and tissue deposited [16,19].

In this study, we quantified variation in development traits (nestling mass and tarsal growth rates, incubation period, and nestling period) across latitude and with a suite of life-history and natural-history traits in a large sample of Nearctic and lowland Neotropical bird species. Our data were gathered primarily from two north temperate sites and one site in lowland Panama. We chose traits that allowed us to assess trade-offs between development time and parental investment strategies, or that are associated with fast or slow phenotypes (Table 1). For instance, longer incubation periods and slower growth are often related to larger adult body mass [36,37] and egg mass [38]. Regional differences in clutch size are often associated with growth rate, with larger clutches being associated with faster nestling growth [37]. We summarize previously observed differences in life-history traits, and predicted correlations between our subset of traits, in Table 1.

## Materials and methods

### Study sites

We collected data at two lowland temperate sites (Michigan and Oregon, USA) and one lowland tropical site (Colon province, Republic of Panama), from 2003–2006.

**Table 1. Observed differences in traits of temperate and tropical passerines, and the predicted relationships with mass and tarsus growth rates ($K$).**

| Variables | | Mass growth rate | Tarsus growth rate | Nestling period | Incubation period |
|---|---|---|---|---|---|
| Mass growth rates | | -- | ↑ | ↓ | ↓ |
| Tarsus growth rates | | ↑ | -- | ↓ | ↓ |
| Nestling period | | ↓ | ↓ | -- | ↑ |
| Incubation period | | ↓ | ↓ | ↑ | -- |
| Clutch size | | ↑ | ↑ | - | ↓ |
| Relative egg mass | | ↓ | ↓ | ↑ | ↑ |
| Adult mass | | ↓ | ↓ | ↑ | ↑ |
| Nest type | open-cup | ↑ | ↑ | ↑ | ↑ |
| | enclosed-cup | ↓ | ↓ | ↓ | ↓ |
| | cavity | ↓ | ↓ | ↓ | ↓ |
| Nest height | | ↓ | ↓ | ↑ | ↑ |
| Incubation type | uni | ↓ | ↓ | - | ↓ |
| | bi | ↑ | ↑ | - | ↑ |
| Parental care type | uni | ↓ | ↓ | ↓ | ↓ |
| | bi | ↑ | ↑ | ↑ | ↓ |

Cells are shaded to indicate predicted positive (dark gray, ↑), negative (light gray, ↓), or neutral (medium gray, -) relationships between the development traits of interest (columns) and a set of associated variables (rows).

[4,7,15,31,33,36–46].

**Michigan (42˚N 85˚W).** Fieldwork was conducted at Lux Arbor Reserve and Kellogg Biological Station (KBS) in southwestern Michigan. Lux Arbor is a 1323-ha managed reserve consisting of agricultural fields, conifer plantations, mixed deciduous forest, wetlands, and meadows bordering a large shallow lake. Annual precipitation averages 89 cm and mean annual temperature is 9.7˚C (http://lter.kbs.msu.edu/). KBS is a landscaped parkland habitat located 17-km from Lux Arbor. Data collection occurred from May to August. In Michigan and Oregon, we installed artificial nest boxes to collect data on secondary-cavity nesting species, (i.e., *Troglodytes aedon*, *Poecile atricapilla*, *Tachycineta* spp., and *Sialia* spp.).

**Oregon (44˚N 123˚W).** We collected additional data on secondary-cavity nesting birds in rural Benton County, Oregon. Boxes were placed on public and private lands including pasturelands, active organic farms, golf courses, and oak savannah within a 24-km radius of Corvallis, Oregon. Average annual rainfall is 103 cm and average annual temperature is 11.5˚C (http://www.ocs.oregonstate.edu/). We installed boxes (without predator guards) on posts, trees, and telephone poles. Data collection occurred between April and early September.

**Panama (9˚N 79˚W).** Fieldwork occurred within, or on the outskirts of, Soberania National Park (NP) (22,000 hectares) in central Panama. Soberania NP is approximately 30-km north of Panama City at the confluence of the Chagres River and the Panama Canal. Our field sites consisted of lowland second growth rainforest as well as parkland habitats in suburban Gamboa. Average rainfall is 260 cm [44], and average annual temperate is about 25˚C [Smithsonian unpublished data; 47,48]. We did not provide nest boxes. Data were collected from March through July annually.

## Field methods

We conducted extensive nest searching in Michigan and Panama. Nests were monitored every 3 days, weather permitting, throughout the breeding cycle until a nest failed or its offspring fledged. During key transition times (laying, hatching, and fledging), we monitored nests every day to ensure accurate estimates of the lengths of incubation and nestling periods. For each species, we recorded egg mass (g), clutch size (#), incubation period (d), nestling period (d), nestling growth rate ($d^{-1}$), daily nest mortality rate (DMR, proportion/day), and nest height (m) above the ground. We also categorized nest type and parental care strategy during both the incubation and the nestling periods for all species. Not all measurements were obtained for all nests because some nests were discovered after nest initiation and nests were not equally accessible.

**Life-history and natural-history variables.** We measured fresh egg mass (± 0.1 g). Because eggs lose 10–15% of their mass over incubation, we also measured their length and width, and for eggs found after clutch completion, we used relationships proposed by Deeming et al. [49] to estimate fresh egg mass from these linear measurements. These estimates were then combined with fresh egg mass to calculate a species' average. We recorded clutch size as the mean number of eggs per clutch. Incubation period was quantified as the time (d) from clutch completion to the hatch of the last chick [50,51]. The nestling period was the time (d) from the hatching of the first nestling to the fledging of the first nestling. We estimated nest height in the field as the distance (m) from the nest to the ground. Adult mass (g) for each species was obtained from published sources [52–54]. For species exhibiting sexual size dimorphism, we averaged male and female masses.

**Daily nest Mortality Rate (DMR).** DMR was calculated using the method of Mayfield [55,56] for species with samples of at least 15 nests. For species with fewer than 15 nests, we calculated DMR using [35]: DMR = -ln($S$)/ $t$, where $S$ is proportion of nests that survived to fledging and $t$ is the length of time (days) that nests held contents. We pooled data within

study sites across years to generate one DMR estimate per species and site. For species with smaller samples of nests, we supplemented our data with values from the literature.

**Categorical variables.**   We included a variable describing region (temperate or tropical). For analyses with incubation period, we included the variable *incubation type*, which was determined from our observations and the literature as either uniparental (one parent predominately incubates the clutch) or biparental (both parents incubate). The variable *parental care* was used in all analyses that involved development during the nestling period. We quantified this trait from our observations and the literature as either uniparental (only one parent contributes to nestling care in the form of brooding or feeding offspring) or biparental (both parents care for young). Nest type was defined as open-cup, enclosed-cup, or cavity/burrow [10].

**Nestling growth.**   We quantified the rate of growth of individually-marked nestlings by measuring changes in mass (± 0.1 g), tarsus length (± 0.1 mm), bill length (± 0.1 mm), and the unflattened wing chord (± 0.5 mm) over time. We individually marked each nestling by coloring a metatarsus with non-toxic felt pen; we weighed nestlings to the nearest 0.1 g (Acculab PocketPro 60 g Electronic Balance; Salter Brecknell Electronic Pocket Balance). We then measured tarsus length (from the metatarsal notch to the opened pad of the foot) and bill length (from the distal end of the nares to the tip of the bill) with electronic calipers (Mitutoyo Digimatic). We measured wing chord with a wing ruler. These data were recorded every 1–4 days until the age at which risk of causing premature fledging became prominent. We supplemented our growth data with data from previously published and unpublished sources (R.E. R.), which we reanalyzed following Austin *et al.* [23]. Data from wrentits (*C. fasciata*) were collected at the Point Reyes National Seashore in Marin County, California [for details see 57] and were provided by Point Blue Conservation Science (https://www.pointblue.org/).

Not all nests were discovered prior to hatch. In such cases, we used morphometrics of known-age nestlings and their growth curves to estimate ages of nestlings. We also assigned ages to developmental milestones, including first eye-opening, approximate total primary feather lengths (5 mm categories), and feather sheath condition (pin or broken sheath). For each developmental milestone, we assigned ages by determining when 50% of individuals in a sample exhibited the trait. We then used these milestones as a rough indicator of nestling maturity [58], and to confirm models of predicted age.

**Growth rates.**   To quantify nestling growth rates for mass and tarsus length, we used the fixed *A* (asymptote) method from Austin et al. [23]. This method accurately characterizes growth and generates unbiased estimates of the growth rate constant *k* [23]. Briefly, we fitted untransformed mass and tarsus measurements to the logistic growth equation for each species (PROC NLIN; SAS Institute, Cary, North Carolina, v9.1–9.3). The formula for logistic growth is $M_t = A/(1 + e^{-k(t-i)})$, where *t* is time (days); $M_t$ is mass (grams) at age *t*; *A* is the asymptote set at the adult morphometric value or, in species where nestling measurements exceed adult values, at the mean peak nestling value; *i* is the inflection point where $M_t = A/2$; and $k$ ($t^{-1}$) is the growth rate constant [23,59]. We then bootstrapped the raw growth data (sampling rate, *n* = 1000; replicates = 1000; PROC SURVEYSELECT) and estimated growth rates for each replicate by species (PROC NLIN). We pooled the estimated parameters (PROC UNIVARIATE) to generate unbiased estimates of error. For more details on age and growth rate estimation, see Austin-Bythell [60]. Data can be found here: https://doi.org/10.7267/m613n4425.

## Statistical analyses

**Multiple linear regression and model selection/averaging.**   We compared mass growth rate, tarsus growth rate, incubation period, and nestling period individually between regions

(PROC MIXED). We also compared these development traits between passerine suborders (Tyranni or Passeri) after accounting for regional and size differences (PROC MIXED). We then conducted model selection (PROC GLMSELECT) to relate individual development traits (*y*, or response variables) to reproductive life-history and natural-history traits (*x*, or explanatory variables). Model selection coupled with model averaging allowed us to simplify the interpretation of the top regression models, and, by bootstrapping models, allowed for only the model with the highest frequency score (relativized by the number of bootstraps), or model weight ($\pi_i$), to be used (ModelAverage, selection = stepwise, nsample = 10,000 iterations, Subset(best = 1); [61]. The top model represents the set of variables that best fit the response variable and have the highest model weight. We then conducted multiple linear regressions on all of the final models, which we present here. We included the following variables in the analyses: mass growth rate, tarsus growth rate, incubation period, nestling period, clutch size, nest height, nest type, parental care strategy during the incubation (Incubation Type) and nestling (Parent Type) periods, DMR, egg mass, and adult mass. We also included the variable *region* to account for differences in daylength between North America and Panama. Including *region* also accounted for any systematic variation related to latitude among our life-history variables. We did not include tarsus growth in regression models that assessed mass growth because these traits are not independent, i.e., they were measured on the same birds as mass growth. Because development traits vary with latitude and organism size, and because we were interested in assessing associations with our variables of interest independent of these variables, we conducted another set of model selections, in which we accounted for *region* and *adult mass* in the final models (by forcing their inclusion).

We ensured that we met model assumptions (normal distribution and uniform variance) for the analyses and determined that the variables were not highly correlated (i.e., variance inflation factor, VIF $\geq$ 10; $R^2 \geq$ 0.7). We transformed all continuous variables to their natural logs (ln) to approximate normal distributions and to improve the fits of the models, with the exception of nest height, which required a square-root transformation. High correlation between *x*-variables increases model variance and can cause spurious results; thus, when two variables were highly correlated ($r > 0.7$, Pearson (continuous) or Spearman (categorical) coefficient), the redundant variable (i.e., the explanatory variable with lower correlation to the response variable) was removed from the model. In this analysis, clutch size was redundant with region ($r = 0.84$) and egg mass was redundant with adult mass ($r = 0.94$). While not technically redundant, tarsus growth rate and incubation period were highly correlated ($r = 0.66$). All response variables, except nestling period, were compared to clutch size and egg mass; for nestling period, adult mass was considered more appropriate. When we forced the inclusion of region and adult mass, we removed clutch size and egg mass from the models to prevent high VIF. Which of the redundant variables was used to compare the different responses is largely irrelevant, and the choice had little effect on the interpretation of the results because both variables are accounted for in the model by the single retained variable. Consequently, we interpret both retained and redundant variables in the results and discussion.

**Discriminant Function Analysis (DFA).** We also explored differences between passerine suborders (Passeri and Tyranni) using a discriminant function analysis of all development traits (PROC DISCRIM).

**Phylogenetic comparison.** We downloaded 2500 phylogenetic trees from birdtree.org [62] ("Stage2_MayrAll_Hackett") and used them to create a maximum clade credibility tree in TreeAnnotator [63], which we then trimmed to include only the species in our analysis. The phylogenies at birdtree.org do not include the subspecies *Troglodytes aedon inquietus*, so we manually added that taxon to the final tree. We did this by using the function "bind.tip" in the R package phytools [64] to add a node connecting *T. aedon aedon* and *T. aedon inquietus* at 2

mya; the two subspecies differ by 4% with respect to their cytochrome *b* nucleotide sequences [65]. We used the resulting tree to optimize Pagel's lambda (a measure of phylogenetic signal) [66] for the residuals of each model and to estimate model parameters, following Revell [67]. These analyses used the gls() function from the package nlme [68] and the corPagel() function from the package ape [69] in R v.3.4.0 (R Core Team, 2017). In three of the standard models, the optimizer failed to converge on an estimate for Pagel's lambda. For all models, we present the *p*-values from the pGLS models following the conventional results. We also ran pGLS on the models determined by model averaging, but the optimizer converged on an estimate for Pagel's lambda for only one model (constancy as a function of nest type, DMR, clutch size, and nest height); the log likelihoods of the other models all increased as Pagel's lambda approached zero.

**Ethics.**   All animal research was conducted under approval of the Oregon State University Institutional Animal Use and Care Committee permit #3011. No endangered or threatened species were handled. All study sites were publicly accessible.

## Results

Across 92 species, mass growth rate varied from $K = 0.161$ $d^{-1}$ (*Elaenia flavogaster*) to $0.754$ $d^{-1}$ (*Geothlypis trichas*). Tarsus growth varied from $K = 0.201$ $d^{-1}$ (*Manacus vitellinus*) to $0.534$ $d^{-1}$ (*Spizella pusilla*) across 49 species. Species with the lowest mass growth rates inhabit the low-land tropics, whereas the highest growth rates occurred in species inhabiting north temperate regions. Tarsi also grew more slowly in tropical birds. Incubation periods were significantly longer in tropical compared to temperate regions, ranging from 10.9 (*Calcarius lapponicus*) to 22.5 d (*Onychorhynchus mexicanus*) in a sample of 151 species. Nestling period overlapped significantly across regions (species = 153), and was not strongly tied to latitude ([Fig 1]). Indeed, the shortest and longest nestling periods (7.4 and 28.5 d) belonged to temperate species (*Calcarius lapponicus* and *Progne subis*, respectively).

When suborder, i.e., Passeri (oscines) versus Tyranni (suboscines), was considered, suboscines generally had lower growth rates and longer incubation periods than oscine passerines, but similar nestling periods. All of the minimum values for mass and tarsus growth rate, and for incubation period, belonged to tropical suboscines, whereas the maximum values belonged to temperate oscines.

### Development trait comparison

We found significant correlations between all pairs of traits ([Table 2]); however, the correlations between nestling period and mass and tarsus growth rates were driven by a few species from families (primarily warblers and sparrows) with rapid growth and short nestling periods ([Fig 2]). Without these species, growth rate and nestling period were not significantly correlated; thus, although nestling period and growth rate are both expressions of development, our results indicate that these are not interchangeable measures. While all the variables are significantly interrelated, correlations between some traits, such as mass and tarsus growth, and tarsus growth rate and incubation period, are stronger than others ([Table 2]).

### Regional analysis

Lowland tropical species in our sample grew, on average, more slowly than temperate species ([Fig 3]). The median mass growth rate was 1.25 times (95% CI: 1.11–1.40) faster in temperate species ($F_{1, 90} = 14.9$, $P < 0.001$, pGLS $P = 0.021$). Median mass growth rates were 0.400 d$^{-1}$ in temperate birds versus 0.318 d$^{-1}$ in tropical birds. Yet, growth rates overlapped substantially between regions. Tarsus growth rates also differed statistically between regions ($F_{1, 47} = 16.0$,

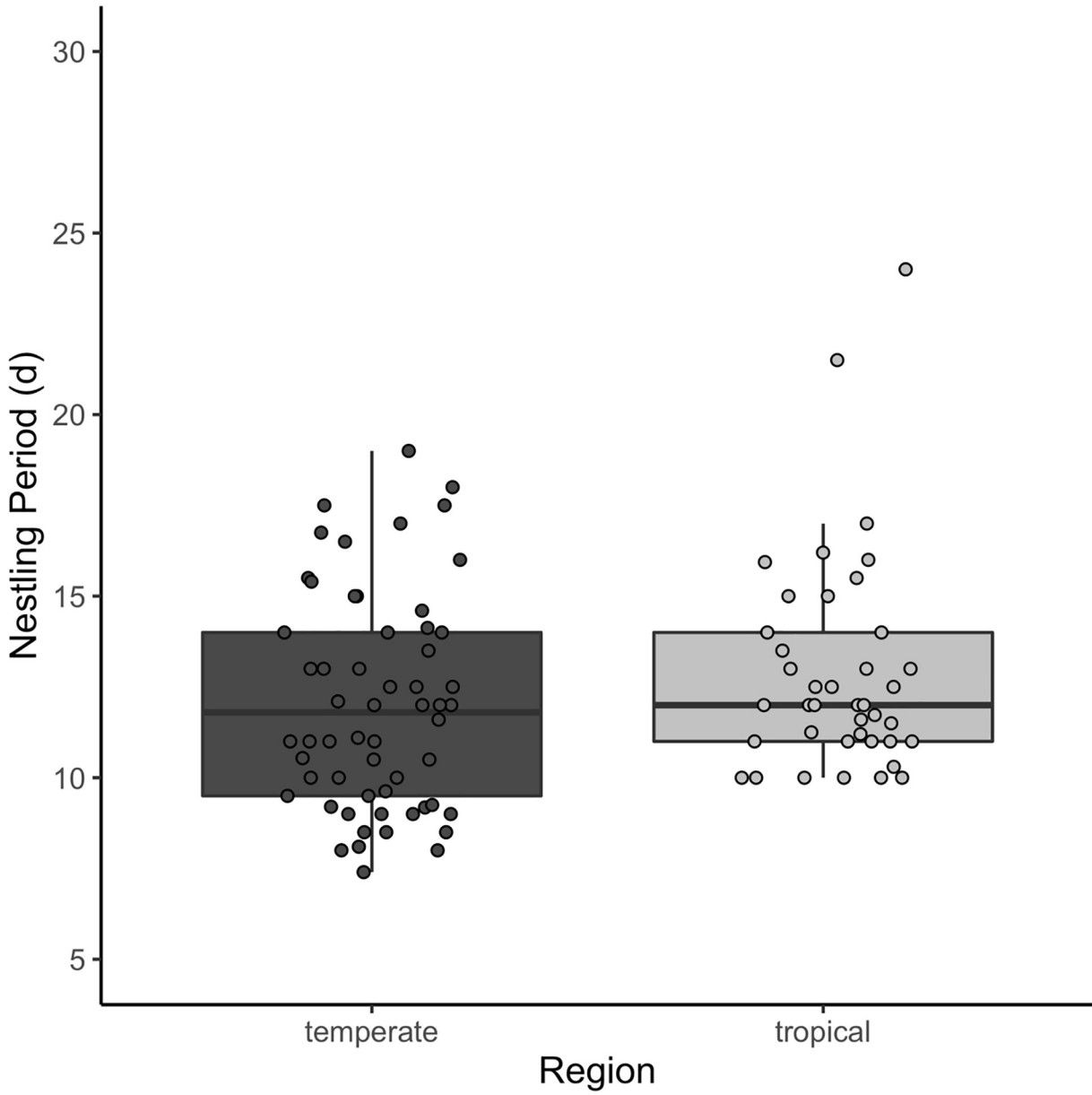

**Fig 1. Nestling period (d) of open-cup nesting species by region.**

$P < 0.001$, pGLS $P < 0.001$), with tropical birds growing 1.26 times (95% CI: 1.12–1.41) slower than temperate species. Temperate birds had a median tarsus growth rate of 0.375 $d^{-1}$ while tropical passerines had a value of 0.313 $d^{-1}$. On average, the incubation periods of tropical songbirds were approximately 1.20 (95% CI: 1.15–1.26; temperate = 13.1 d, tropical = 15.8 d) times longer than those of temperate species ($F_{1, 149} = 68.6$, $P < 0.001$, pGLS $P < 0.001$). Nestling periods differed significantly between regions, as well ($F_{1, 151} = 6.2$, $P = 0.014$; pGLS $P = 0.008$), but this result largely reflects the fewer cavity nesting species in the tropics (region $F_{1, 149} = 2.3$, $P < 0.133$, pGLS $P = 0.052$, nest type $F_{2, 149} = 38.6$, $P < 0.001$; pGLS $P < 0.001$). When we included only open-cup nesting species in the analysis, we found that there was no

**Table 2. Covariation of development traits for the subset of passerines included in this study.**

| Pearson correlation coefficients: | | | | |
|---|---|---|---|---|
| **Variables** | **Mass Growth rate ($d^{-1}$)** | **Tarsus Growth rate ($d^{1}$)** | **Nestling period (d)** | **Incubation period (d)** |
| **Mass growth rate ($d^{-1}$)** | | $n = 49$ | $n = 92$ | $n = 92$ |
| | | 0.65 | -0.31 | -0.39 |
| | | $P < 0.001$ | $P = 0.003$ | $P < 0.001$ |
| **Tarsus growth rate ($d^{1}$)** | | | $n = 49$ | $n = 49$ |
| | | | -0.55 | -0.77 |
| | | | $P < 0.001$ | $P < 0.001$ |
| **Nestling period (d)** | | | | $n = 148$ |
| | | | | 0.51 |
| | | | | $P < 0.001$ |

regional difference in nestling period ($F_{1, 96} = 2.9$, $P = 0.095$); however, when phylogeny was accounted for, there appeared to be a regional difference (pGLS $P = 0.008$).

## Suborder analyses

**Discriminant function analysis.** Suboscines typically had longer incubation periods, and lower mass and tarsus growth rates, than oscine passerines (model Wilks' lamba = 0.48, $F_{4, 44} = 11.8$, $P < 0.001$; growth rate, $F_{1, 47} = 12.8$, $P < 0.001$, pGLS $P = 0.739$; tarsus growth rate, $F_{1, 47} = 17.3$, $P < 0.001$, pGLS $P = 0.346$; incubation period, $F_{1, 47} = 34.7$, $P < 0.001$, pGLS $P = 0.093$). The length of the nestling period did not differ between suborders after variation in the development traits was accounted for ($F_{1, 47} = 1.2$, $P = 0.28$, pGLS $P = 0.90$).

**Linear models.** Both mass and tarsus growth rate were higher in the suborder Passeri, even after accounting for differences in latitude and adult size (mass growth rate: suborder: $F_{1, 88} = 8.3$, lsmeans difference (values have not been back-transformed) Passeri vs. Tyranni = 0.16 ± 0.06 s.e., $P = 0.005$, pGLS $P = 0.454$, region $F_{1, 88} = 12.6$, lsmeans difference temperate vs. tropical = 0.18 ± 0.05 s.e., $P < 0.001$, pGLS $P = 0.005$, adult mass $F_{1, 88} = 42.0$, est. = -0.24 ± 0.04 s.e., $P < 0.001$, pGLS $P < 0.001$; tarsus growth rate: suborder: $F_{1, 45} = 14.9$, lsmeans difference = 0.20 ± 0.05 s.e., $P < 0.001$, pGLS $P = 0.812$, region $F_{1, 45} = 15.1$, lsmeans difference = 0.19 ± 0.05 s.e., $P < 0.001$, pGLS $P < 0.001$; adult tarsus length $F_{1, 45} = 7.0$, est. = -0.24 ± 0.09 s.e., $P = 0.011$, pGLS $P = 0.407$). Incubation period was shorter among oscines than among suboscines after accounting for region and mass differences (suborder: $F_{1, 148} = 46.4$, lsmeans difference = -0.17 ± 0.02 s.e., $P < 0.001$, pGLS $P = 0.133$, region $F_{1, 148} = 26.6$, lsmeans difference = -0.12 ± 0.02 s.e., $P < 0.001$, pGLS $P < 0.001$, adult mass $F_{1, 148} = 1.8$, est = 0.02 ± 0.02 s.e., $P = 0.183$, pGLS $P = 0.007$). In contrast to the discriminant function analysis, nestling period was shorter in oscines than suboscines, though with substantial overlap (suborder: $F_{1, 150} = 7.4$, lsmeans difference = -0.15 ± 0.05 s.e., $P = 0.007$, pGLS $P = 0.77$; region $F_{1, 150} = 1.3$, lsmeans difference = -0.06 ± 0.05 s.e., $P = 0.26$, pGLS $P = 0.011$; adult mass $F_{1, 150} = 7.2$, est = 0.09 ± 0.03 s.e., $P = 0.008$, pGLS $P < 0.001$). The difference in results may have been caused by methodological differences; however, unlike the DFA, the linear model did not take into account variation in the other development traits. As such, the significant contribution of suborder detected in the linear model may have resulted from a suborder difference that was unaccounted for in the overall pattern of slow growth, and which is better explained by mass and tarsus growth rates and incubation period. As before, when nest type was added as an explanatory variable, it accounted for most of the variation in the model ($F_{2, 147} = 42.4$, $P < 0.001$, open-cup: pGLS $P < 0.001$, enclosed-cup pGLS $P = 0.955$) although suborder ($F_{1,}$

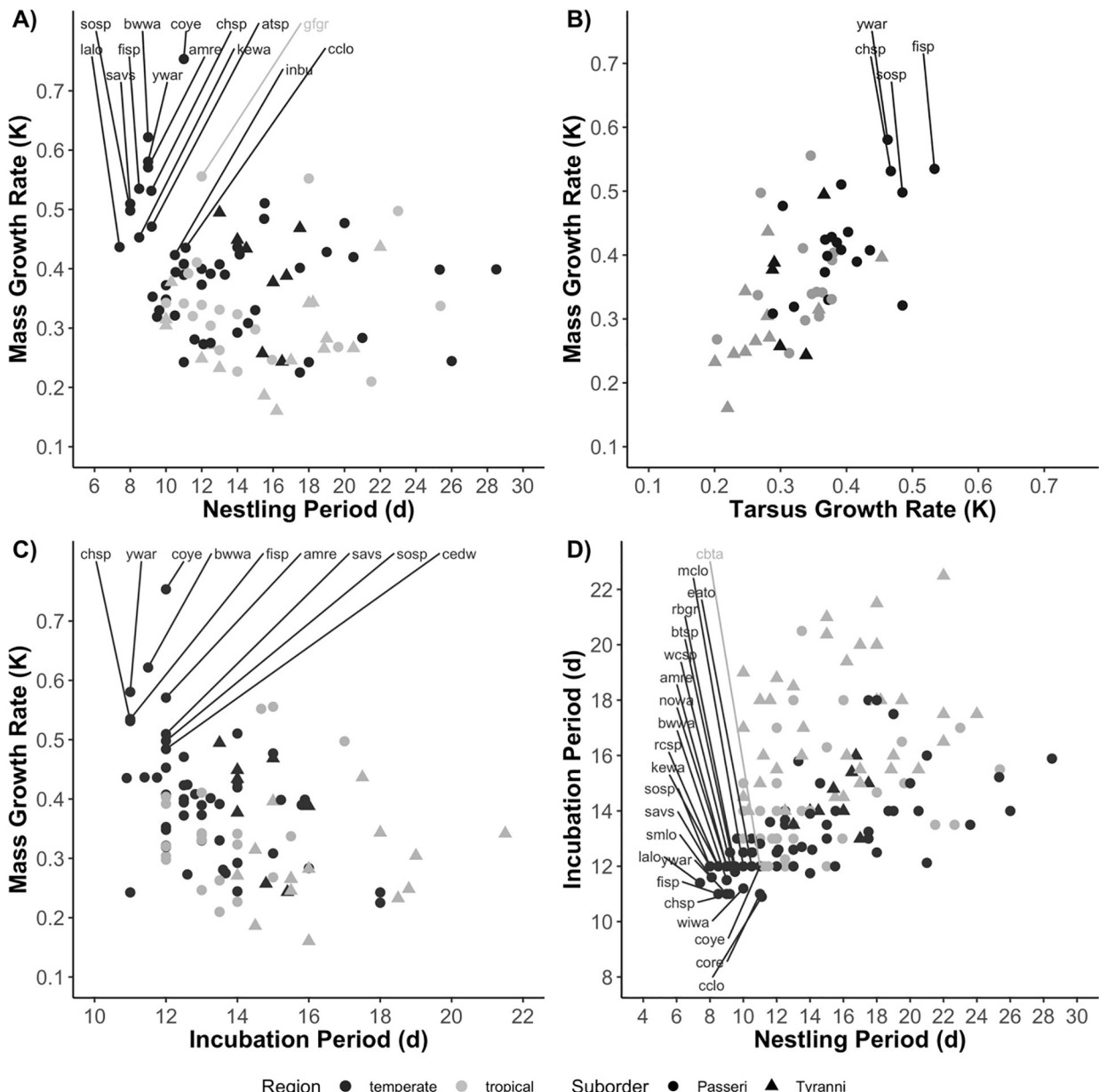

**Fig 2.** Relationships between mass growth rate and (A) nestling period, (B) tarsus growth rate (*k*), and (C) incubation period. Plot (D) depicts the relationship between incubation period and nestling period. Each point represents a particular species. Latitudinal regions are distinguished by dark gray (temperate species) or light gray (tropical), while Suborders are indicated by circles (Passeri or oscines) and triangles (Tyranni or suboscines). We have highlighted several groups (sparrows, longspurs, and warblers) that seem to be driving some of the observed associations. Species are labeled by their 4 letter alpha codes (see https://doi.org/10.7267/m613n4425). Sparrow species include atsp (American tree sparrow), chsp (chipping sparrow), eato (eastern towhee), fisp (field sparrow), savs (savannah sparrow), rcsp (rufous-collared sparrow), and wcsp (white-crowned sparrow). Longspurs include cclo (chestnut-collared longspur), lalo (lapland longspur), and mclo (McCown's longspur). Warblers are amre (American redstart), bwwa (blue-winged warbler), coye (common yellowthroat), kewa (Kentucky warbler), nowa (northern waterthrush), wiwa (Wilson's warbler), and ywar (yellow warbler).

$_{147} = 7.1$, $P = 0.009$, pGLS $P = 0.71$) and adult mass ($F_{1, 147} = 13.5$, $P < 0.001$, pGLS $P = 0.001$) remained statistically significant. Region was not related to nestling period after nest type was included in the analysis ($F_{1, 147} = 0.0$, $P = 0.870$, pGLS $P = 0.078$).

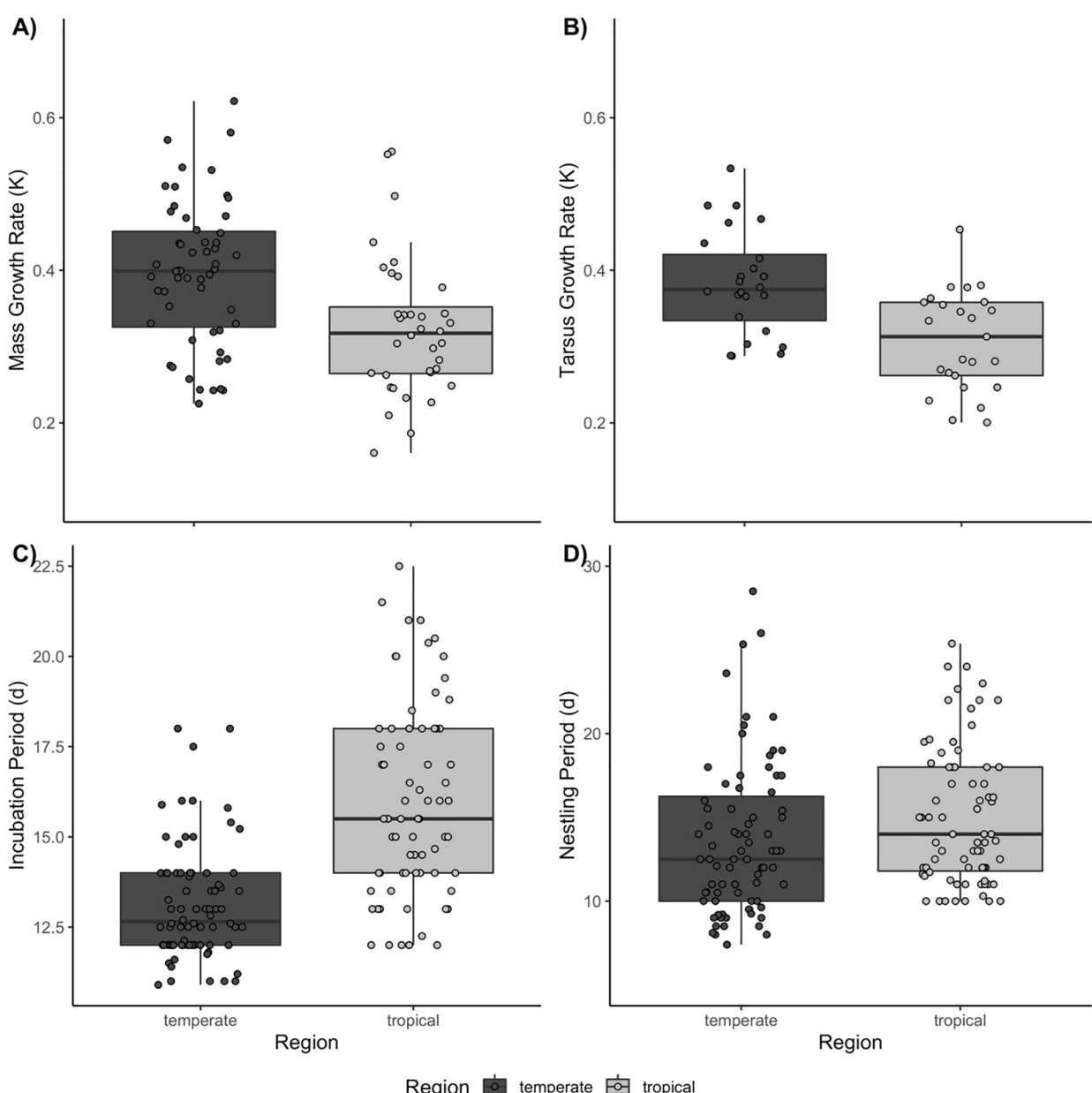

**Fig 3.** Development traits by region A) Mass Growth Rate *k*, B) Tarsus Growth Rate, *k*, C) Incubation Period, d, D) Nestling Period, d.

### Life-history relationships

**Mass growth.** *In the top model for mass growth ($\pi_i = 0.172$), growth rate was positively correlated with clutch size ($F_{1, 86} = 8.0$, est. = $0.22 \pm 0.08$ s.e., $P = 0.006$, pGLS $P < 0.001$). Our top model indicated that mass growth was negatively correlated with incubation period, nestling period, and egg mass (incubation period $F_{1, 86} = 3.2$, est. = $-0.41 \pm 0.23$ s.e., $P = 0.079$, pGLS $P = 0.117$; nestling period $F_{1, 86} = 3.4$, est. = $-0.19 \pm 0.11$ s.e., $P = 0.067$, pGLS $P = 0.361$; egg mass $F_{1, 86} = 24.0$, est. = $-0.25 \pm 0.05$ s.e., $P < 0.001$, pGLS $P < 0.001$). As such, slower postnatal growth was correlated with longer incubation periods. There was also a suggestive positive relationship between mass growth and nestling period. Finally, species with large egg mass*

(and adult mass as a redundant variable) had slower mass growth rates. When we controlled for region and adult mass (by forcing their inclusion in all models), we found that, as before, mass growth rate was correlated with region, adult mass (redundant variable = egg mass), and incubation period ($\pi_i$ = 0.116; region $F_{1, 86}$ = 9.2, lsmeans difference: tropical–temperate = -0.146 ± 0.048 s.e., $P$ = 0.003, pGLS $P$ = 0.004; adult mass $F_{1, 86}$ = 38.9, est. = -0.211 ± 0.034 s.e., $P < 0.001$, pGLS $P < 0.001$). Controlling for region and adult mass did not change the direction of any of the aforementioned relationships. It did indicate that in our sample of species, the lowland tropical birds had an overall pattern of slow growth compared to the temperate species. Nest type was also included in the top model (nest type $F_{2, 86}$ = 3.3, $P$ = 0.043). This relationship was primarily driven by the differences in mass growth rate between open-cup and cavity nesting species (lsmeans difference: open-cup–cavity = -0.153 ± 0.061 s.e., $P$ = 0.014, enclosed-cup–cavity = -0.076 ± 0.091 s.e., $P$ = 0.41), pGLS nest type: open-cup $P < 0.001$, enclosed-cup $P$ = 0.32).Open-cup nesting species tended to have faster mass growth than cavity nesting species.

**Tarsus growth.** In the top model ($\pi_i$ = 0.131) for tarsus growth, growth rate was positively correlated with clutch size ($F_{1, 45}$ = 3.6, est. = 0.10 ± 0.05 s.e., $P$ = 0.066, pGLS $P$ = 0.084). This result indicates that species with fast tarsus growth rate also tended to have larger clutches though this suggestive relationship was not significant. We also found that tarsus growth rate negatively correlated with incubation period ($F_{1, 45}$ = 73.4, est. = -1.14 ± 0.13, $P < 0.001$, pGLS $P < 0.001$) and nest height ($F_{1, 45}$ = 12.0, est. = -0.12 ± 0.03 s.e., $P$ = 0.001, pGLS $P$ = 0.009). Species with slow tarsus growth tended to have long incubation periods and nested higher from the ground. When region and size (adult tarsus) were accounted for in the model, we found that the top model ($\pi_i$ = 0.153), tropical birds had slower tarsus growth rate in our sample of birds, as expected, ($F_{1, 44}$ = 9.0, lsmeans difference: temperate–tropical = -0.117 ± 0.039, $P$ = 0.004, pGLS $P$ = 0.004). Tarsus growth rate was not correlated to the mature length of the tarsus (tarsus length $F_{1, 44}$ = 0.1, est. = -0.027 ± 0.074 s.e., $P < 0.72$, pGLS $P$ = 0.49). As with the top model, for which we did not enforce regional and size differences, incubation period and nest height were still related to tarsus growth rate. Specifically, slower growth rates were associated with longer incubation periods ($F_{1, 44}$ = 51.6, est. = -1.04 ± 0.145 s.e., $P < 0.001$, pGLS $P < 0.001$) and higher nest heights ($F_{1, 44}$ = 12.9, est. = -0.118 ± 0.033 s.e., $P < 0.001$, pGLS $P$ = 0.007).

**Incubation period.** The incubation period top model ($\pi_i$ = 0.135) included tarsus growth ($F_{1, 45}$ = 49.9, est. = -0.44 ± 0.06 s.e., $P < 0.001$, pGLS $P$ = 0.013), nestling period ($F_{1, 45}$ = 7.2, est. = 0.14 ± 0.05 s.e., $P$ = 0.010, pGLS $P$ = 0.006), and nest height ($F_{1, 45}$ = 8.9, est. = -0.07 ± 0.03 s.e., $P$ = 0.005, pGLS $P$ = 0.007). We found that incubation period was negatively related to tarsus growth rate, meaning that birds with long incubation periods also tended to have slow tarsus growth. Incubation period was also positively correlated with nestling period, suggesting that birds with longer incubation periods tended to have longer nestling periods. Long incubation periods were also correlated with lower nest height, which is an artifact of nest type. When nest type was included in all models, nest height was no longer significantly correlated with incubation period; only tarsus growth rate was related to incubation period length ($\pi_i$ = 0.12). After forcing the inclusion of region and adult mass in the top model, in order to account for their effects, we found that region was not significantly related to incubation period after accounting for differences in adult mass effects ($\pi_i$ = 0.092; region $F_{1, 45}$ = 0.1; region lsmeans difference: temperate–tropical est. = -0.006 ± 0.106 s.e., $P$ = 0.827, pGLS $P$ = 0.168). Meanwhile, species with larger masses tended to have longer incubation periods ($F_{1, 45}$ = 7.3, est. = -0.059 ± 0.022 s.e., $P$ = 0.010, pGLS $P$ = 0.093). The only other variable that was retained in this model was tarsus growth rate ($F_{1, 45}$ = 55.4, est. = -0.462 ± 0.062 s.e., $P < 0.001$, pGLS $P$ = 0.006), which was negatively related to incubation length.

**Nestling period.** Top models for nestling period ($\pi_i = 0.148$) included nest type [nest type $F_{2, 66} = 8.2$, $P < 0.001$ (lsmeans differences: open-cup–cavity = -0.20 ± 0.06, $P = 0.001$, pGLS $P = 0.033$; enclosed-cup–cavity = -0.03 ± 0.08 s.e., $P = 0.700$, pGLS $P = 0.517$)], DMR ($F_{1, 66} = 24.0$, est = -0.15 ± 0.03 s.e., $P < 0.001$, pGLS $P < 0.001$), nest height ($F_{1, 66} = 38.5$, est = 0.17 ± 0.028 s.e., $P < 0.001$, pGLS $P < 0.001$), mass growth rate ($F_{1, 66} = 21.5$, est = -0.33 ± 0.07 s.e., $P < 0.001$, pGLS $P < 0.001$), and egg mass (redundant variable = adult mass; $F_{1, 66} = 3.3$, est = -0.08 ± 0.044 s.e., $P = 0.073$, pGLS $P = 0.136$). Nestling period was significantly shorter in species with open-cup nest structure vs. either enclosed-cup or cavity nests.

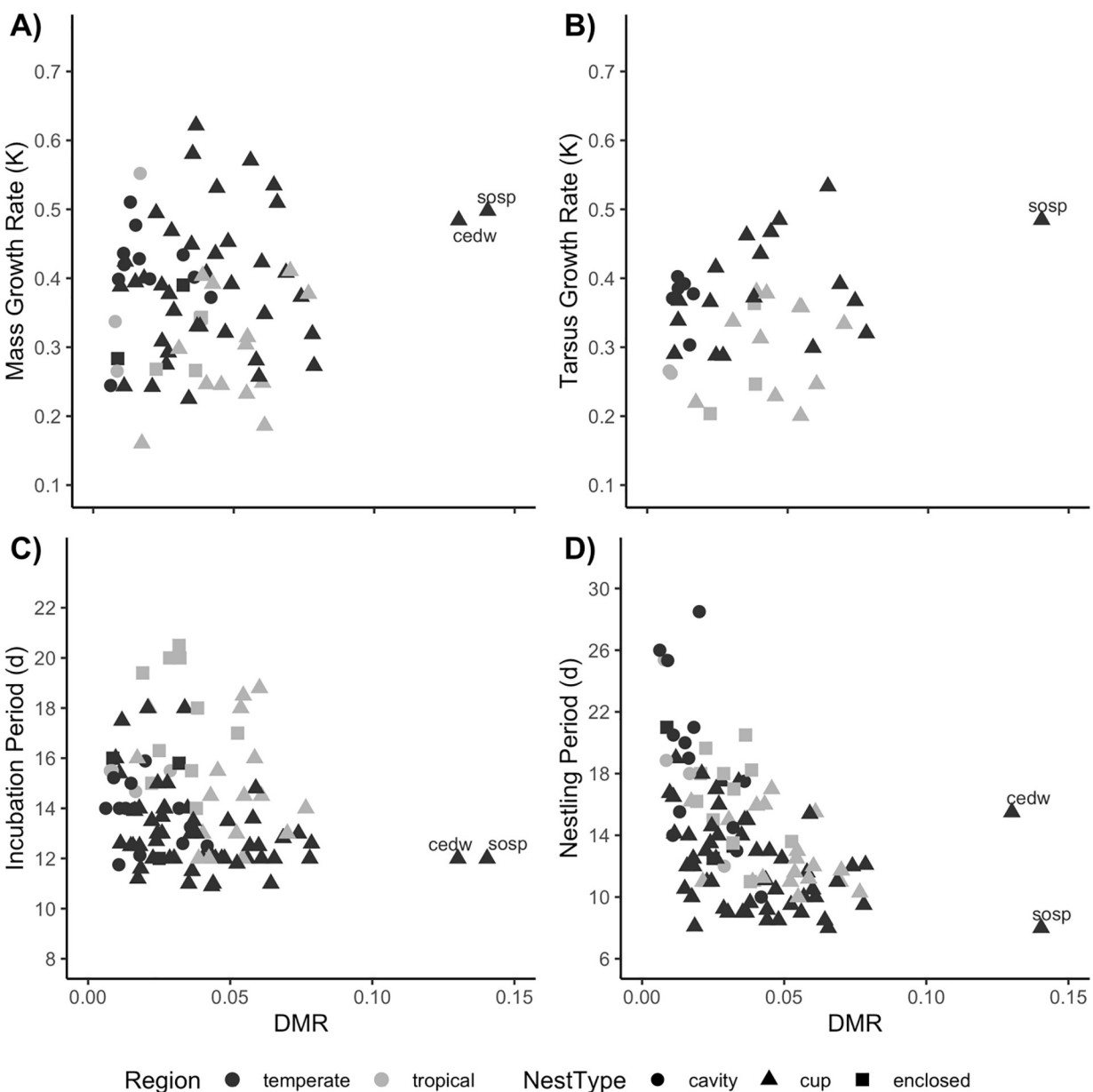

**Fig 4.** Relationships between DMR on the *x*-axis and A) Mass Growth Rate, *k*; B) Tarsus Growth Rate, *k*; C) Incubation Period, d; or D) Nestling Period, d, on the *y*-axis. Each point represents a different species. Temperate species are depicted in dark gray and tropical species in light gray, while nest types are indicated by circles (cavity), triangles (open-cup), or squares (enclosed-cup). Outliers with high DMR are two temperate open-cup nesting species, *Bombycilla cedrorum* (cedw) and *Melospiza melodia* (sosp).

Species with higher nest mortality also tended to have shorter nestling periods (Fig 4) and nest closer to the ground. Longer nestling periods were correlated with higher mass growth rates. Finally, nestling periods were suggestively (not significantly) negatively related to egg/adult mass. After forcing region and adult mass into the top model, we found a similar result as before. Nestling period was correlated with nest type, DMR, mass growth rate and nest height ($\pi_i = 0.677$; region $F_{1,65} = 0.0$, (temperate–tropical est. = 0.002 ± 0.046 s.e.), $P = 0.97$, pGLS $P = 0.249$; adult mass $F_{1,65} = 0.6$, est. = -0.026 ± 0.034 s.e., $P = 0.44$, pGLS $P = 0.407$; nest type $F_{2,65} = 9.0$, (lsmeans difference: open-cup–cavity est. = -0.216 ± 0.058 s.e., adj. $P < 0.001$, enclosed-cup–open-cup est. = -0.025 ± 0.079 s.e., adj. $P = 0.93$), $P < 0.001$, pGLS: open-cup $P = 0.061$, enclosed-cup $P = 0.54$; DMR $F_{1,65} = 21.9$, est. = -0.149 ± 0.032 s.e., $P < 0.001$, pGLS $P < 0.001$; mass growth rate $F_{1, 65} = 13.5$, est. = -0.298 ± 0.081 s.e., $P < 0.001$, pGLS $P = 0.007$; nest height $F_{1, 65} = 35.5$, est. = 0.169 ± 0.028 s.e., $P < 0.001$, pGLS $P < 0.001$].

## Discussion

Embryonic and post-embryonic growth was 20–26% slower, on average, in our sample of lowland tropical bird species than in our sample of temperate bird species. The 49 and 153 temperate and tropical species, respectively, provide new perspectives on relationships among a larger suite of life-history traits than in past studies, as well as more precise measurements of these traits for some commonly studied species. Although our data come primarily from two temperate sites and one tropical location, they include more than 160 species from a wide range of family-level taxa. The extent to which our species represent entire tropical and temperate bird faunas will remain unclear until more data can be gathered across the western hemisphere. Nevertheless, our data are consistent with earlier observations of a latitudinal gradient in development rate [7,21,35,70] and show some counter-intuitive associations between traits. Nestling periods, for example, did not, on average, differ statistically between temperate and tropical songbirds, after accounting for nest type. The longer nestling periods of secondary cavity nesting species among the temperate species seem to have driven the overall correlation between region and nestling period. Thus, we confirm both the difference in nestling period related to nest type and the limited variation in this trait in relation to latitude.

We found a significant negative correlation between growth rate and nestling period ($F_{1,90} = 0.6$, $P = 0.003$); however, the correlation was very weak (adjusted $R^2 = 0.09$) and the trend was driven by several north temperate species with very short nestling periods ($< 9.5$ d) and rapid growth ($> 0.400$ d$^{-1}$) from the families Parulidae (New World warblers), Calcariidae (longspurs), and Passerellidae (New World sparrows); without these species, no growth rate-nestling period correlation appears (Fig 2). Development rate varied between passerine suborders, with Tyranni (suboscines) having slower growth and longer development periods than Passeri (oscines), after accounting for differences in region and adult size. We found that mass and tarsus growth rates, and incubation period, were related to suborder and region. Nestling period differed by suborder but not region, after accounting for well-known differences in nest type. Once variation in nest type was included in the model, we found that Passeri had shorter nestling periods than Tyranni.

Similar to mass growth, tarsus growth also was slower in tropical passerines. This suggests that an underlying constraint limits the growth rate of lowland tropical, compared to temperate, passerines. It may be the case that mass growth rate is limited by the growth of the most constrained tissues [19]. Starck [71] suggested that the limiting tissue may be the long bones, however, later studies produced conflicting results [72]. Long bone growth is largely dictated by the size of the cartilaginous proliferation zone and the level of ossification of the bone at hatching in altricial nestlings; this tends to vary widely among species [71,72]. Considering the

significantly slower growth in tropical birds, yet their similar nestling periods to temperate birds (after differences in nest type were accounted for), our result suggests that tropical birds fledge at a smaller relative size and lesser state of development compared to temperate species. Perhaps increased post-fledging parental care in the tropics [11] helps to compensate for the lower relative size and developmental maturity of nestlings at fledging.

Comparisons of developmental traits with other life-history characteristics generated conflicting results, indicating that selection may be acting differently on these traits. However, general syndromes did appear. For instance, mass growth rate was positively correlated with clutch size (and, thus, region) and negatively related to incubation period, nestling period, and adult (and egg) mass. Slowly growing species tended to have large eggs (and hence large adult masses), long incubation periods, and long nestling periods. Tarsus growth rate was also positively correlated to clutch size (region) and negatively correlated to incubation period. One key difference between mass growth rate and tarsus growth rate was the inclusion of nest height in the tarsus growth model. Species that nested low to the ground tended to have more rapid tarsus growth. Species with low nests also tended to have shorter nestling periods, likely due to the increased predation pressure that species with ground-level or low-level nests experience [10].

Incubation period also was negatively related to tarsus growth, nest height, and nestling period. These variables appear to be linked, as they were also included as key variables when tarsus growth was the dependent variable. These results suggest either that tarsus growth and incubation period are intrinsically linked, or that selective pressure on these traits is similar. For example, sparrows nest close to the ground, have short incubation and nestling periods, and rapid tarsus growth. Perhaps their terrestrial lifestyle and foraging style select for rapid growth. While one might expect that nest predation imposes selection on these traits, we found no correlation between DMR and mass or tarsus growth and incubation period. Because nestling period is more labile than the more physiologically constrained rate of growth, we hypothesized that, to reduce overall mortality, the nestling period would be statistically negatively associated with variation in DMR, which it was. Species with shorter nestling periods tended to have higher DMR, build open-cup nests, and nest close to the ground. They also had faster mass growth and lower adult and egg mass, but some of these trends were driven by the same group of birds (e.g., north temperate sparrows, warblers, and longspurs) that influenced the earlier correlation between growth rate and nestling period (Fig 3). When we accounted for regional and morphometric differences in the models, results were similar, with a few exceptions. Both tarsus growth rate and nestling period produced the same models apart from statistically redundant variables related to either region or adult mass. Mass growth rate was correlated with nest type and incubation period, but not nestling period, though some variation in nestling period may have been accounted for in the nest type variable, as the two are associated. Incubation period was related to tarsus growth rate only when region and adult mass were included, which suggests the variation associated with nestling period and nest height was somehow related to regional or morphometric differences.

Selection should favor shorter nestling periods when nest predation is high, as is common in the tropics [this study; 4,10,21,28]. Selection applied to nestling growth rate by time-dependent nest mortality is not as clear or direct as is selection on the nestling period itself, and it is constrained by physiological and phylogenetic considerations [as discussed in 4,19,23,36]. Results from our comparative analyses, where only nestling period correlated with daily nest mortality rate, corroborate the limited selective influence of DMR on growth rate compared to its effect on nestling period in lowland songbirds. That is to say, high rates of time-dependent mortality are associated with shorter nestling periods but have little effect on incubation period

length or nestling growth rate. In contrast, other studies have found that nestling growth rate is correlated with time-dependent nest mortality [21,22].

The lack of a correlation between nestling growth rate and nestling period in this study can be attributed to our use of unbiased measures of nestling growth rate that are independent of the effects of nestling period on the growth trajectory [see discussion in 23]. Early fledging and attainment of a transient growth plateau below adult mass can potentially bias estimates of growth rates to higher values. For example, Remeš and Martin [22] and Martin [21] found that growth rates of passerine birds varied in direct relation to nestling period and daily nestling mortality rates, suggesting that postnatal growth rate increases in response to selection on the length of the vulnerable period, *contra* Ricklefs's [4,18,31,73] suggestion that postnatal growth is pushed by even weak selection to a physiological limit inversely related to tissue maturity. Early fledging is often associated with a more linear trajectory of growth near fledging or with the development of a transient growth plateau, which, when used to estimate the asymptote of the growth curve, inflates estimates of the growth rate. Thus, growth rate might be confounded with the conspicuous response of the length of the nestling period to time-dependent mortality [22,23]. Fitting models to a floating asymptote, where the trajectory of the growth curve is linear or incomplete at fledging, biases the growth rate ($k$), generally inversely to the length of the nestling period. However, when the complete growth curve is used to estimate $k$, or the asymptote is set to the adult value, the growth rate is less biased and, typically, assumes a lower value. Austin et al. [23] discussed this effect at length and simulated how incomplete growth curves can influence growth rate estimates. Such biased estimates of growth rates affect our understanding of trait associations in comparative analyses [23]. Care must also be taken in the types of morphological measures that are used to estimate growth rates. Traits that are incompletely developed at fledging should not be used to estimate growth rates owing to the error inherent in these estimates, which depend on estimated asymptotes. For instance, wing chord length is often ~ 50% of the adult size in passerines at fledging [unpublished data; 21]. Fitting a growth curve to such data, even with a fixed asymptote, can inflate growth rate estimates, and, potentially, produce spurious correlations in downstream analyses.

Incubation periods, too, differ between temperate and tropical passerines [29,30,70,74–77]. The paradox, that incubation periods are longer in the tropics in spite of higher time-dependent mortality, suggests that parents attempt to lower their personal risk at the nest site, and nest predation more generally, by decreasing nest attendance. This is thought to extend the length of the incubation period, but substantial disagreement exists in the literature. The lack of consensus may reflect differences in field sites (lowland vs. highland, different latitudes, etc.). While temperature clearly plays a significant role in determining the duration of incubation, our work in the lowland tropics, including the use of artificial incubation, has found that natural fluctuations in egg temperature and adult attendance during incubation do not explain variation in the length of the incubation period [29,30,70,75–77]. Rather, intrinsic constraints appear to determine the embryo development periods of lowland tropical birds. Here, we find additional evidence that time-dependent nest mortality does not explain the longer incubation periods of lowland tropical birds.

Instead, we are persuaded that the outcome of selection on embryonic and post-embryonic growth rates reflects optimized strategies that balance the conflicting demands of parents and nestlings [30,31]. Nestlings are constrained by physiological limits to growth rate [36]. Parents influence growth through their ability to provide resources (i.e., food, heat, protection from predators) and in determining the number of offspring in a brood (thereby, affecting the degree of sibling competition). Parents are also constrained by their own needs for energy, self-maintenance activities, and safety from predators. Thus, parents must also optimize the balance between their investment in the current brood and their investment in future

reproduction and survival [78]. Because offspring success influences parental fitness, these conflicting demands likely reflect an optimized investment strategy that minimizes deleterious effects on both nestlings and parents under average conditions.

Much research on avian life histories has focused on temperate-tropical contrasts in the expression of life-history phenotypes. Our data from a highly diverse sample of species continue to support general differences in growth rate (both mass and tarsus) and incubation period between tropical (slower) and temperate (more rapid) birds. Differences in life histories between regions provide insight into how the ecological characteristics of each region interact with the physiological limits of passerines to shape life histories. While there are clear latitudinal differences in the expression of development traits, many of these traits also overlap extensively between regions. Of the many traits that vary across latitude, growth rates are among the most prominent. Slow embryonic and post-embryonic growth are related to a general slow pace-of-life syndrome in tropical species, but other environmental factors (e.g., photoperiod length, food limitation) and intrinsic factors (e.g., immune function, bone growth, metabolism) may contribute in ways that have yet to be understood.

## Acknowledgments

We thank administrators of the Smithsonian Tropical Research Institute (Panama), Kellogg Biological Station (Michigan State University), and Oregon State University for allowing us to conduct research at their facilities and for providing logistical support. All animal research was conducted under Oregon State University Institutional Animal Use and Care Committee permit #3011. We are grateful to Dennis Jongsomjit and Point Blue Conservation Science for allowing us to use unpublished wrentit data. Thanks to Daniel Roby for helpful comments on earlier drafts. We gratefully acknowledge assistance in the field from: J. Y. Adkins, A. Battin, D. W. Bradley, J. R. Bruce, N. Chartier, R. L. Gamboa, J. Junda, L. Miller, B. L. Perez, N. K. Strycker, and R. E. Zambrano.

## Author Contributions

**Conceptualization:** W. Douglas Robinson, Tara Rodden Robinson, Robert E. Ricklefs.

**Data curation:** Suzanne H. Austin.

**Formal analysis:** Suzanne H. Austin, Vincenzo A. Ellis.

**Funding acquisition:** W. Douglas Robinson, Robert E. Ricklefs.

**Investigation:** Suzanne H. Austin, W. Douglas Robinson, Tara Rodden Robinson.

**Methodology:** Suzanne H. Austin, W. Douglas Robinson, Tara Rodden Robinson, Robert E. Ricklefs.

**Project administration:** Robert E. Ricklefs.

**Supervision:** W. Douglas Robinson, Robert E. Ricklefs.

**Writing – original draft:** Suzanne H. Austin.

**Writing – review & editing:** Suzanne H. Austin, W. Douglas Robinson, Tara Rodden Robinson, Vincenzo A. Ellis, Robert E. Ricklefs.

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
