## [Decision Letter · Decision Letter 0]

30 Dec 2019

PONE-D-19-31241

Development of New World temperate and tropical songbirds

PLOS ONE

Dear Dr. Austin,

Thank you for submitting your manuscript to PLOS ONE. After careful consideration, we feel that it has merit but does not fully meet PLOS ONE’s publication criteria as it currently stands. Therefore, we invite you to submit a revised version of the manuscript that addresses the points raised during the review process.

Please see the attached reviews.  Reviewer. 1 had no major criticisms, but reviewer 2 (Dr. Sherry) raised a number of issues that need attention.  I highlight especially his comments about the reliance entirely on one site in Panama as "tropical" when in fact it may differ in many ways from other tropical sites.  I don't think this invalidates the study, but these site-related limitations need to be acknowledged and dealt with more directly, and the limitations of using a single site from Panama should temper some of the sweeping conclusions about temperate versus tropical species.  Please also address his other concerns and comments in your revision.

We would appreciate receiving your revised manuscript by Feb 10 2020 11:59PM. To enhance the reproducibility of your results, we recommend that if applicable you deposit your laboratory protocols in protocols.io, where a protocol can be assigned its own identifier (DOI) such that it can be cited independently in the future. For instructions see: http://journals.plos.org/plosone/s/submission-guidelines#loc-laboratory-protocols

We look forward to receiving your revised manuscript.

Sincerely,

Charles R. Brown.

Academic Editor

PLOS ONE

Journal Requirements:

2. Thank you for including your ethics statement: All animal research conducted under IACUC permit #3011.

a. Please amend your current ethics statement to include the full name of the ethics committee that approved your specific study.

For additional information about PLOS ONE submissions requirements for ethics oversight of animal work, please refer to http://journals.plos.org/plosone/s/submission-guidelines#loc-animal-research  

'Project funding was provided by National Science Foundation IRCEB grant #0212587 to RER and WDR. VAE was supported by a postdoctoral fellowship from the Carl Tryggers Foundation.'

'Project funding was provided by National Science Foundation IRCEB grant #0212587 to RER and WDR.'

Please provide an amended Funding Statement that declares *all* the funding or sources of support received during this specific study (whether external or internal to your organization) as detailed online in our guide for authors at http://journals.plos.org/plosone/s/submit-nowPlease state what role the funders took in the study.  If any authors received a salary from any of your funders, please state which authors and which funder. If the funders had no role, please state: "The funders had no role in study design, data collection and analysis, decision to publish, or preparation of the manuscript."

5. Please amend either the title on the online submission form (via Edit Submission) or the title in the manuscript so that they are identical.

6. Please include captions for your Supporting Information files at the end of your manuscript, and update any in-text citations to match accordingly. Please see our Supporting Information guidelines for more information: http://journals.plos.org/plosone/s/supporting-information

7. Your ethics statement must appear in the Methods section of your manuscript. If your ethics statement is written in any section besides the Methods, please move it to the Methods section and delete it from any other section. Please also ensure that your ethics statement is included in your manuscript, as the ethics section of your online submission will not be published alongside your manuscript.

Reviewers' comments:

Reviewer's Responses to Questions

**Comments to the Author**

1. Is the manuscript technically sound, and do the data support the conclusions?

Reviewer #1: Yes

Reviewer #2: Yes

2. Has the statistical analysis been performed appropriately and rigorously? 

Reviewer #1: Yes

Reviewer #2: I Don't Know

3. Have the authors made all data underlying the findings in their manuscript fully available?

Reviewer #1: Yes

Reviewer #2: Yes

4. Is the manuscript presented in an intelligible fashion and written in standard English?

Reviewer #1: Yes

Reviewer #2: Yes

5. Review Comments to the Author

Reviewer #1: Although my own research has only dealt with a few smaller aspects of the many topics studied here, I think I know enough about what is going on with the various comparisons about fast and slow patterns across regions and habitats to provide a good review of this paper. And this is a very impressive paper, both because of the size of the data set analyzed and the background of the authors. Although I always find model selection papers hard to analyze in detail, that is the nature of the beast. But this analysis seems to be quite complete and very clear, with an impressive sample size that covers many regions and types of birds. The discussion does an excellent job of summarizing the results and it should provide a long list of hypotheses that require further testing as the authors and others build on this data set.

Reviewer #2: This study includes what the authors justly claim is a “large suite” of life-history traits, which provide novel and invaluable insights into avian development and life history traits in general (interrelatedness of traits) and latitudinal comparisons, tropical vs. temperate, in particular. The suite of multiple traits makes the comparisons of the impacts of various ecological factors particularly valuable and convincing. For example, the importance of nest site (and relative safety thereof) also comes through clearly, particularly in the tropical species. One novelty this study points out is how different the incubation stage is from the nestling stage latitudinally, i.e., there’s an interaction between stage of the nesting cycle and latitude with respect to life-history traits.

The difference between oscine and suboscine life-histories in the tropics is also really important, particularly considering the recent influential life-history study by Martin (Science, reference 20), which included no suboscine passerines, a serious problem when treating the Neotropics.

An important strength, but also a limitation, of this study is its tropical analyses restricted to the one Panamanian site, Colon Province. The strength is that all the tropical comparisons come from the same (Panamanian) site, studied over a long time period, thus controlling for some of the kinds of factors that influence life histories. The weakness is that Panama does not represent the tropics, or even the Neotropics, which are diverse. Diversity within the tropics is well known, including sites with very different rainfall patterns, soils, geography, etc., and montane sites that of course differ with respect to life-histories. However, the Panamanian sites do not even represent lowland wet Neotropics. Some of these authors (specifically reference 44, Robinson et al. 2000) describe how their Panamanian study site compared to sites in South America has significantly reduced species richness, far fewer rare species, and many more migratory species seasonally, among many other differences in the bird communities. To the extent that life-history adaptations are density-dependent (as Ricklefs has argued), we should expect that Panamanian life-histories may not represent South American ones. We also have to wonder at how representative the temperate sites are, but at least here multiple study sites far removed from each other entered the data set. This cautionary argument does not invalidate the value of this study at all—these are really valuable data, extremely hard to come by, and revealing, but this problem of representativeness at least needs acknowledgement. The authors thus mislead by stating throughout the manuscript that this study is a tropical-temperate comparison, because their N’s are basically 1 tropical and 2 temperate sites. This is pure pseudoreplication.

Interestingly, this manuscript points out that several oscine passerine groups have anomalously short incubation and nestling periods, and rapid growth rates. These species include sparrows, longspurs, and warblers, all emberizoids, and they are mostly (but not exclusively) temperate. These species likely entered the Americas via Beringia, and some of them colonized the American tropics, all very recently compared to the suboscine passerines that have been evolving in the Neotropics for tens of millions of years, at least (see time-constrained passerine phylogeny by Oliveros et al., PNAS 116, pp. 7916-7925, 2019). The ancestor of these emberizoid birds was likely migratory, and thus all these species may retain biased temperate-to-north-temperate life-histories, depending on how long it takes for these complex suites of life-history traits to evolve—maybe a long time. I think it’s worth considering at least that some of the most interesting results of this study reflect evolutionary history (inertia?). This should at least be mentioned.

Much of the literature on tropical-temperate comparisons of avian life-history evolution has focused on nest predation. High tropical nest predation rates, particularly for open cup-nesting birds, is well documented, starting with Skutch’s particularly influential work based on his extraordinary natural history observations over a long period in Costa Rica. It’s relatively simple to document high predation rates in these nests, particularly low nests. Less is known about high nests, and more difficult-to-observe hole nests and pendant nests, but information has been accumulating. This manuscript intimates that more may be going on with life histories than just the impacts of nest predation (lines 555-558): “Rather, intrinsic constraints appear tolimit the embryo development periods of lowland tropical birds. Here, we find additionalevidence that time-dependent nest mortality does not explain the longer incubation periods oflowland tropical birds.” I could not agree more with this statement. My own research in the tropics indicates that food availability is a very important, and vastly underappreciated factor affecting life-histories. I cannot prove this (yet; I’m working on manuscripts that make this argument), and I’m not suggesting any major re-working of the manuscript in this context. However, I would suggest at least another sentence making explicit how little we know about the diets and potential food-limitation of tropical birds. The long incubation periods of many suboscine passerines are particularly interesting in this context, and these have never been adequately explained. I will cite this manuscript the moment it gets published because it lays out nicely where we need better explanations and more research.

Some more minor comments/corrections:

Reference numbering issues (e.g., reference 22 numbered 23).

Fig. 1. Species codes are indecipherable on figures due to the size-reduction of these figures.

Results, line 325: “slower” not “faster”?

Section of Results “Regional Analysis” does not explicitly cite Fig. 2, as it should. Similarly, the second of two Fig. 3’s not cited in section “Nestling period”. Two Figure 3’s is, of course, a confusing problem itself.

Fig. 2 subfigures not labeled a-d to correspond with Fig. 2 legend.

Inconsistent upper/lower case usage in subheadings within Results.

Discussion, lines 452-463: Since Parulidae, for sure, and sparrows additionally (enough species for comparisons) included both temperate and tropical species, and plenty of lowland tropical and temperate Parulidae, these comparisons within family will be really pivotal and interesting in the future. By restricting this present study opportunistically to the Panama species, a lot of this important variation was missed.

Discussion lines 471-474: This seems speculative. Is there some independent way to assess these relative development rate differences? Do tropical birds, for example, fledge at lower relative body mass compared to adults?

Discussion line 480: Should be “(and egg) mass. …”

Line 486 needs period at end of sentence = end of line.

Starting at least by line 510 in Discussion, some reference numbers to citations are incorrect. E.g., in lines 513 and 523 reference 23 should be 22. In line 524, reference numbers are clearly incorrect. This is frustrating because at this point I am not sure which references back up which assertions.

Table 1 provides a clever way to compile a large number of observed (from past literature), and thus predicted relationships between growth rates and lengths of incubation and nestling periods with a variety of ecological and life-history traits. However, I found the shading to be non-intuitive, and confusing. Might this table work better with a single line in each box, with negative slope for negative relationship, positive slope for positive relationship, and horizontal line for no relationship? Also, I was confused by some of these relationships. For example, nest type is an important predictor of developmental traits, and the positive correlations (dark shading) with open nests with mass and tarsus growth rates makes sense, but positive correlations of open nests and length of incubation and nestling periods do not make sense, or am I missing something?

Results were difficult to read in places. For example, in the Life-history Relationships section, Mass growth subsection, the parentheses were not matching (some were missing?), and this section was almost undecipherable. Nest type had multiple comparisons among three different types, all jumbled together in one sentence. Shorter sentences would help.

The many errors I’ve identified indicate a degree of sloppiness in preparing this manuscript that need to be corrected by very careful copy editing by the authors.

6. PLOS authors have the option to publish the peer review history of their article (what does this mean?). If published, this will include your full peer review and any attached files.

Reviewer #1: No

Reviewer #2: Yes: Tom (Thomas) W. Sherry

---

## [Author Response · Author response to Decision Letter 0]

18 Apr 2020

27 March 2020

Dear Dr. Brown

We appreciate the comments of the two reviewers and have edited our paper accordingly to improve its quality.

Reviewer 2, Dr. Sherry, offered several thoughtful suggestions and caught some typographical errors, the latter of which we believe to have corrected.

We strongly agree with Dr. Sherry that the literature in this area has a common problem of over-extending the reach of results by claiming that data from a small set of species represents “tropical birds” or “temperate species.” This is an issue that has frustrated us as well; Dr. Sherry raised the example of studies that entirely lack suboscines, a dominant group of tropical birds. Although we tried to avoid that same problem, we were reminded that sometimes we over-extended our reach, so we have edited the paper in several places to make clear that we are analyzing and discussing “our sample or set of lowland tropical and temperate birds.” We trust it will be even clearer to readers now that they are aware our data are from Panama and from two temperate sites and might not be representative of tropical or temperate localities generally. Astute readers know that data in these kinds of studies originate from particular localities, and incorporate that knowledge into their interpretation of results. We did appreciate that both reviewers recognized what a massive amount of work it required for us to assemble these data for more than 150+ species. 

Dr. Sherry’s idea about mentioning food limitation is a good one. We now raise this issue in a couple of places, particularly lines 109-110 and later in the discussion, although briefly. This, along with the idea of invasion of temperate species via Beringia, are worthy ideas but in this already long and complicated manuscript, we felt it best to leave these ideas for better development elsewhere. We are aware that Dr. Sherry is working on a book where he might be able to flesh out these concepts more fully than we could here.

In addition to fixing a few typos, we have adjusted Figure 1, 2, and 4 to indicate which panels are A-D, and improved the font sizes on the species codes so they are more visible.

We have double-checked the references, especially in the Discussion, where Dr. Sherry recognized that something had gone wrong and some needed repair.

We also revised Table 1 to make it more easily interpretable.

We hope that our efforts have improved the readability of the paper and appreciate the helpful advice from the reviewers.

Please let us know if you have any questions.

Sincerely,

Suzanne Austin

---

## [Decision Letter · Decision Letter 1]

11 May 2020

Development syndromes in New World temperate and tropical songbirds

PONE-D-19-31241R1

Dear Dr. Austin,

We are pleased to inform you that your manuscript has been judged scientifically suitable for publication and will be formally accepted for publication once it complies with all outstanding technical requirements. Dr. Sherry reviewed your revision, and was satisfied that you adequately addressed his concerns, and I concur.

Sincerely,

Charles R. Brown

Academic Editor

PLOS ONE

Additional Editor Comments (optional):

Reviewers' comments:

Reviewer's Responses to Questions

**Comments to the Author**

1. If the authors have adequately addressed your comments raised in a previous round of review and you feel that this manuscript is now acceptable for publication, you may indicate that here to bypass the “Comments to the Author” section, enter your conflict of interest statement in the “Confidential to Editor” section, and submit your "Accept" recommendation.

Reviewer #2: All comments have been addressed

2. Is the manuscript technically sound, and do the data support the conclusions?

Reviewer #2: Yes

3. Has the statistical analysis been performed appropriately and rigorously? 

Reviewer #2: Yes

4. Have the authors made all data underlying the findings in their manuscript fully available?

Reviewer #2: Yes

5. Is the manuscript presented in an intelligible fashion and written in standard English?

Reviewer #2: Yes

6. Review Comments to the Author

Reviewer #2: All the issues I identified in the first review have been addressed in this version of the manuscript, so I find the manuscript much improved.

The one exception is that the authors did not (in the Discussion) get into possible historical/taxonomic explanations for some of the patterns, which is fair enough. I can accept that this gets into issues beyond the data presented here, and represents a kind of analysis that will need to await new data and new analyses dedicated to the topic.

This manuscript represents an invaluable new set of analyses on developmental rates and durations in relation to other life history variables (like egg size, clutch size, nest survival rate) and ecological factors like nest type and height. This manuscript points out a lot of misconceptions, or oversimplifications in the literature to date, and adds a lot of new patterns, analyses, and findings. This manuscript reinforces a lot of patterns that are already well documented, giving some degree of assurance that these trends and patterns are quite general. There is much that is new here, making this a valuable manuscript for publication.

7. PLOS authors have the option to publish the peer review history of their article (what does this mean?). If published, this will include your full peer review and any attached files.

Reviewer #2: Yes: Thomas W. Sherry

---

## [Editor Report · Acceptance letter]

5 Aug 2020

PONE-D-19-31241R1 

Development syndromes in New World temperate and tropical songbirds 

Dear Dr. Austin:

I'm pleased to inform you that your manuscript has been deemed suitable for publication in PLOS ONE. Congratulations! Your manuscript is now with our production department. 

Kind regards, 

on behalf of

Dr. Charles R. Brown 

Academic Editor

PLOS ONE